# Phase I Targeted Combination Trial of Sorafenib and GW5074 in Patients with Advanced Refractory Solid Tumors

**DOI:** 10.3390/jcm11082183

**Published:** 2022-04-14

**Authors:** Chien-Chang Kao, Ching-Liang Ho, Ming-Hsin Yang, Yi-Ta Tsai, Shu-Yu Liu, Ping-Ying Chang, Yi-Ying Wu, Jia-Hong Chen, Tzu-Chuan Huang, Ren-Hua Yehn, Ming-Shen Dai, Yeu-Chin Chen, Guang-Huan Sun, Tai-Lung Cha

**Affiliations:** 1Division of Urology, Department of Surgery, Tri-Service General Hospital, National Defense Medical Center, Taipei 11490, Taiwan; guman2011@gmail.com (C.-C.K.); msy681028@gmail.com (M.-H.Y.); ghsun1@gmail.com (G.-H.S.); 2Graduate Institute of Medical Sciences, National Defense Medical Center, Taipei 11490, Taiwan; feg81230@gmail.com (Y.-T.T.); ayufish.candy@gmail.com (S.-Y.L.); 3Division of Hematology/Oncology, Department of Medicine, Tri-Service General Hospital, National Defense Medical Center, Taipei 11490, Taiwan; hochingliang@yahoo.com.tw (C.-L.H.); max-chang@yahoo.com.tw (P.-Y.C.); rq0922@gmail.com (Y.-Y.W.); ndmc_tw.tw@yahoo.com.tw (J.-H.C.); ralph0909@gmail.com (T.-C.H.); minceing1202@gmail.com (R.-H.Y.); dms1201@gmail.com (M.-S.D.); yeuchin99@gmail.com (Y.-C.C.)

**Keywords:** DAPK, sorafenib, GW5074, advanced tumor

## Abstract

Background: Combination therapy with the administration of GW5074 and sorafenib significantly induced necrotic death in various cancer cells in vivo, as well as prolonging the survival of an animal disease model due to significant suppression of the primary and metastatic lesions. We sought to determine the safety, tolerability, pharmacokinetics, and anti-tumor activity of this co-administration therapy in patients with refractory advanced solid cancers. Methods: Twelve patients were enrolled. Eligible subjects received different dosages of GW5074 in one of the three dose cohorts (Cohort 1: 750 mg daily, Cohort 2: 1500 mg daily, Cohort 3: 750 mg twice daily) plus 200 mg of sorafenib daily to determine the maximum tolerated dose (MTD) and dose limiting toxicities (DLT) at phase 1. Furthermore, the expression level of phosphorylated DAPK^S308^ in primary tumor, metastatic tumor, and circulating tumor cells (CTC) were evaluated to investigate the relationship between biomarker and the efficacy profile. Results: Among the 12 enrolled patients in this phase 1 trial, most adverse effects (AE) were grade 1, with two being grade 3. The most frequent AE of all grades were weight loss and hypertension, occurring in 16.7% of participants. Eight patients (66.7%) had the disease controlled by receiving co-administration therapy of GW5074 and sorafenib. GW5074 was found to have poor absorption, as increasing the dosage did not result in a significant increase in the bioavailability of GW5074 in subjects. Furthermore, the expression level of phosphorylated DAPK^S308^ in tumor and CTCs were correlated with the disease control rate (DCR) and duration of response (DOR). Conclusions: Co-administration therapy of GW5074 and sorafenib demonstrated a favorable safety profile and showed anti-tumor activity in a variety of tumor types. However, the solubility of GW5074 is not satisfactory. A future phase 2a trial will be carried out using the new salted form that has been proven to be more effective.

## 1. Introduction

Resistance to chemotherapy and molecular targeted treatment is a major challenge in current cancer research and treatment. In clinics, there are unmet medical needs in drug discovery to treat cancers that are no longer susceptible to current treatments. The mechanism of resistance includes shared features such as ineffective induction of cell death, activation of pro-survival pathways, and alteration of the drug target [1,2].

Recent studies showed that developing therapeutic strategies that selectively target cancer cell mitochondria as a novel mechanism may bring effective treatments for cancer patients. Death-associated protein kinase (DAPK), a serine/threonine protein kinase, functions as a tumor-suppressor gene, and its expression is attenuated in many cancer types [3,4,5,6]. Previous studies showed that auto-phosphorylation of DAPK at residue Serine 308 contributes to the loss of tumor suppressor function in cancer with DAPK expression [7,8,9]. On the other hand, Raf inhibitors have been shown to have survival benefits for a variety of cancers; however, emerging problems include rapid development of resistance in clinics and the lack of long-term therapeutic efficacy [10,11,12,13]. A previous publication revealed the potential combination therapeutic effect of GW5074, a C-Raf inhibitor, and sorafenib on cancer cells [9]. GW5074 sensitizes cancer cells to sorafenib, which introduces the anti-cancer effect [9]. The interaction of GW5074 and sorafenib induces a conformational change in the N-terminal domain of C-Raf in the presence of pDAPK^S308^, which compromises the mitochondrial-targeting effect of C-Raf. The facilitated translocation of C-Raf/pDAPK^S308^ complexes from the mitochondria to the cytoplasm results in mitochondrial dysfunction, followed by the generation of reactive oxygen species (ROS). The generated ROS facilitates PP2A-mediated dephosphorylation of pDAPK^S308^ and dissociation of the protein phosphatase-2A (PP2A) from the C-Raf/pDAPK^S308^ complex in the cytoplasm, activating the kinase catalytic activity of DAPK. The ROS generation and activated DAPK cause two-hit damage in cancer cells and contribute to profound cell necroptosis [9]. Although GW5074 and sorafenib are both Raf inhibitors, their use in combination provides a novel mechanism of anti-tumor activity targeting cancer cell necroptosis induced by mitochondrial dysfunction.

We performed this open-label clinical trial to investigate the efficacy, safety, and pharmacokinetic (PK) profile of co-administration therapy of GW5074 (MG005) and sorafenib in metastatic cancer patients. Furthermore, due to the poor solubility of MG005, this study created a new salt form version of GW5074 and evaluated the pharmacokinetic (PK) profile in the preclinical settings.

## 2. Material and Methods

### 2.1. Patient Eligibility

The study includes 12 patients evaluated at our hospital. Patients had refractory advanced or metastatic disease which has no effective standard therapy available. Inclusion criteria included: (a) Eastern Cooperative Oncology Group performance status ≤ 2; (b) age > 20 years; (c) patient has at least one measurable lesion according to Response Evaluation Criteria in Solid Tumors (RECIST) version 1.1; (d) at least 2 weeks post any therapeutic modalities prior to initial dosing. Exclusion criteria included: (a) patients having a history of primary malignancy other than the entry diagnosis; (b) gastrointestinal disease that may alter the absorption of GW5074 (Metagone Biotech, Taipei, Taiwan) and sorafenib (Bayer, Taipei, Taiwan); (c) patients with known brain metastasis; (d) patients who are receiving substances that are potent inducers of CYP3A4 activity; and (e) patients who are receiving g sensitive substrates of CYP1A2, 1B1, 2C8, 2C19 and 3A4 with narrow therapeutic windows.

### 2.2. Study Design

This was a dual-agent, open-label, phase I study. Eligible patients received different dosages of GW5074 (MG005) in one of three dose cohorts plus 200 mg of sorafenib. We conducted the trial following the 3 + 3 dose-escalation phase I clinical trial design [14]. Dose cohorts were escalated sequentially from Cohort 1 at monotherapy with 750 mg QD MG005 for 4 weeks, followed by 750 mg QD MG005 plus 200 mg QD sorafenib to Cohort 2 at 1500 mg QD MG005 plus 200 mg QD sorafenib, and Cohort 3 at 750 mg BID MG005 plus 200 mg QD sorafenib. There were three cohorts and at least two patients were enrolled for each dose level. No intra-patient dose escalation was allowed, and each patient’s first dose was at least 5 days apart from the next patient’s first dose. DLTs were assessed during the initial 8-week and 4-week treatment periods for Cohort 1 and Cohorts 2/3, respectively. The criteria for DLTs are listed in Appendix A. Concurrent administration with 200 mg of sorafenib once daily was implemented for each patient. The enrolled patients were treated until disease progression.

The primary endpoints for the current trial were the MTD and DLTs. The secondary endpoints were the safety and tolerability. Efficacy and pharmacokinetic evaluations were also the secondary endpoints.

### 2.3. Pharmacokinetics Assessments

A plasma concentration–time profile for MG005 as a single agent was established, and PK parameters in terms of the following were also determined: peak plasma concentration (C_max_)/at steady state (C_max,ss_); time to reach peak concentration (T_max_)/at steady state (T_max,ss_); AUC in 1 dosing interval at steady state (AUC_0__→τ,ss_).

### 2.4. CTC and Immunofluorescence Staining

For CTC isolation, CELLection™ Pan Mouse IgG Dynabeads^®^ (4.5 μm; Thermo Fisher Scientific Inc., Taipei City, Taiwan) in combination with an anti-EpCAM antibody [Ber-EP4] (ab7504; Abcam, Cambridge, UK) were used. Blood samples were lysed with red blood cell (RBC) lysis buffer. Cells were then incubated with ~1.6 × 10^6^ anti-EpCAM antibody-coated magnetic beads. Cancer cells adhering to beads were retrieved by running the isolation protocol on the IsoFlux™ machine (Fluxion, Alameda, CA, USA). Isolated cancer cells were fixed in 4% PFA and added onto glass slides, on which a circle with the same size as the magnet had been drawn using a water-repellent Dako pen. The glass slide was placed on top of the magnet when adding or removing buffer from the cells. The cells were stained with PE conjugated anti-CD45 [5B-1] antibody (130-080-201; MACS MiltenyiBiotec, San Diego, CA, USA; 1:100), FITC-conjugated anti-CK [CK3-6H5] antibody (130-080-101; MACS MiltenyiBiotec, 1:10), and pDAPK (orb156534; BiorByt, St. Louis, MO, USA). The cells were stained with DAPI to enable the visualization of cell nuclei. The sample was mounted using Dako Faramount aqueous mounting medium.

### 2.5. Immunohistochemistry Staining

Formalin-fixed paraffin-embedded (FFPE) tumor blocks were sectioned (3 μm thick). Antigen retrieval was carried out via microwaving in preheated EDTA buffer. Sections were blocked using horse serum (1:75) and were incubated with the phospo-DAPK antibody (Biorbyt, orb156534, 1:100) overnight at 4 °C. Staining was developed using DAB solution (BioSB, Santa Barbara, CA, USA, BSB 0257S) and counterstained with hematoxylin for 2 min. AnH-score higher than 48.3 is considered positive for pDAPK staining. H-score = (cancer tissue stain range × intensity) − (normal tissue staining range × intensity).

### 2.6. Cell Cultures, Drug Test, and Cell Viability

The ACHN cells were obtained from the Bioresource Collection and Research Centre (BCRC, Taipei, Taiwan). The cells were cultured in culture media as recommended by the ATCC and kept at 37 °C in a 5% CO_2_, 95% humidity incubator. Before the spike-in experiments, cells were harvested using trypsin−EDTA (Sigma-Aldrich, Taipei, Taiwan) and resuspended in RPMI medium. For the drug test, 5000 cells per well were seeded triply in 96-well plates and were treated with different drug combinations. Cell viability was detected by MTT assay as previously described [15].

### 2.7. Pharmacokinetics Study with MG010

Single-dose oral pharmacokinetics investigation was performed with four beagles (12–14 months, 2 male, 2 female, 10.28–11.52 kg). Four of each dog received 11 mg/kg MG010, with a 7-day washout period between each experiment. A total of 2 mL of blood was collected at time 0, 0.25, 0.5, 1, 2, 4, 6, 8, 16, and 24 h after drug administration. Blood samples were obtained via jugular venipuncture and were centrifuged with a clinical centrifuge at 2500× *g* for 10 min; plasma was removed and stored at −80 °C until analysis.

### 2.8. Statistical Analysis

Unless stated otherwise, results are expressed as the means ± SD. Statistical analyses were performed with SPSS V.22.0 or Prism version 7 (GraphPad Software Inc., La Jolla, CA, USA) comparing continuous variables by non-parametrical Mann–Whitney U and Kruskal-Wallis tests. For all tests, *p* values of less than 0.05 were considered significant.

## 3. Results

### 3.1. Participant Characteristics, Dose Escalation, and MTD

From July 2018 through November 2019, 12 participants were enrolled, with three participants each in Cohort 1 and Cohort 2 and six in Cohort 3 (Table 1). The median (range) age was 56.6 (25–68) years; seven (58%) participants were men and five (42%) were women. All of them are Taiwanese. Three, three, and six patients were enrolled in dose levels 1, 2, and 3, respectively. At the first dose level, DLT was not observed. Three patients were enrolled at dose level 2. Although no DLT was observed, pneumonia grade 3 was reported in one of the patients. Six patients were accrued at dose level 3. From six patients, one developed grade 3 subcostal pain, which was less likely to be related to the treatment. As no DLT was encountered at dose level 3, 750 mg BID MG005 plus 200 mg sorafenib continuously given daily was declared the recommended phase 2 dose (RPTD).

### 3.2. Safety Outcomes

A full list of all adverse events (AEs) at least possibly attributed to sorafenib or MG005 is included in Table 2. Most AEs were grade 1 (*n* = 7, 58.3%), with two (16.7%) being grade 3. The most frequent AEs of all grades were weight loss and hypertension, occurring in 16.7% of participants. Gastrointestinal toxicity, which was generally mild, including abdominal pain (16.7%), nausea (8.3%), GERD (8.3%), erosive gastritis (8.3%), duodenal ulcer (8.3%), and vomiting (16.7%) were found. However, no direct correlation was observed between higher doses and incidence of these phenomena. Leukopenia was the most relevant hematological toxicity (1 patient; 8.3%), which is grade 1 and does not require intervention. The represented dermatologic adverse events, previously reported with single-agent administration of sorafenib, included acneiform rash (8.3%), hand–foot syndrome (8.3%), hyperkeratosis (16.7%), lipoma (8.3%), papule (8.3%), and eczema (16.7%); all were tolerable. Another notable toxicity was musculoskeletal AEs, including sacroiliitis (8.3%) and subcostal pain (8.3%). Subcostal pain was defined as a grade 3 AE, but it was more likely to be attributed to complications of progressive cancer. One patient had liver function impairment (8.3%), which was self-limited. Conjunctivitis (8.3%) and gynecomastia (8.3%) were also observed, and no intervention was required. A safety concern was raised by a grade 3 pneumonia, which was reported in a 65-year-old male subject from Cohort 2. He presented with cough, acute bronchitis, and pneumonia, which were due to a chest infection. There were no toxic deaths in this study.

### 3.3. Pharmacokinetic Analysis

Nine patients were eligible for pharmacokinetic (PK) analysis. The PK properties of MG005 and sorafenib were studied after repeated oral administration of 750 mg QD, 1500 QD, and 750 mg BID in combination with sorafenib 200 mg QD for 4 weeks (Table 3). Mean AUC_0→τ,ss_, T_max_, and C_max_ values of MG005 showed significant intra-patient and inter-patient variability. The average exposure of MG005 (i.e., AUC_0→τ,ss_) in patients of Cohort 1 was 5639.9 ± 2675.9 h·ng/mL, with an average C_max_ of 685.6 ± 335.4 ng/L. No significant increase in the average AUC_0→τ,ss_ or C_max_ was observed when the dosage of MG005 was increased to 1500 mg in Cohort 2. This indicates that a dose-dependent increase in the bioavailability of MG005 cannot be achieved with MG005 at doses beyond 750 mg/day. However, between Cohort 2 and Cohort 3, taking MG005 BID increased AUC_0→τ,ss_, which might suggest that increasing the frequency of MG005 treatment can increase the drug concentration in circulation. The PK data of sorafenib for this study showed that the average AUC_0→τ,ss_ and C_max_ in Cohort 2 were similar to the results for sorafenib monotherapy reported previously [16]. However, in Cohort 1, the average AUC_0→τ,ss_ and C_max_ in patients were higher than those in Cohort 2, although the dosage of sorafenib was the same. This showed that a higher dose of MG005 might interfere with the PK and metabolism of sorafenib.

### 3.4. Efficacy Outcomes

Tumor response was assessed in 12 patients. Regarding the efficacy endpoints, no subjects had achieved either complete response (CR) or partial response (PR), with the disease control rate (DCR) being accounted for in the patients who had achieved stable disease (SD) as the best overall response per RECIST. A total of 3/3, 2/3, and 3/6 subjects had SD at the first tumor assessment (Week 8) for Cohort 1, Cohort 2, and Cohort 3, respectively, representing an 8-week DCR of 100%, 66.7%, and 50%, respectively, over the whole population. Among the patients with SD, the maximum changes in the sum of their target lesions ranged from 16.7% shrinkage to 7.1% growth (Figure 1A), and the median duration of treatment was 300, 85, and 154 days in Cohorts 1, 2, and 3, respectively (Figure 1B). For PFS, the median PFS time was not reached for Cohort 1 and was 111.5 days and 126.5 days for Cohort 2 and Cohort 3, respectively (Figure 1C). We also checked the pDAPK+ circulating tumor cells (CTC) in each patient. The mean CTC number was 12.1 ± 4.3 and the percentage of pDAPK+ CTC was 33.3 ± 12.7. Intriguingly, we found that the proportion of pDAPK+ CTC numbers were highly correlated with the treatment duration (Pearson r = 0.629, *p* = 0.028; Figure 1D,E). This result provides the possibility of prediction of prognosis before the patient receives treatment.

### 3.5. DAPK Expression in Primary vs. Metastatic Tumor

The expression levels of pDAPK^S308^ were found to be higher when compared with their matched normal counterparts [9]. Moreover, our previous results showed that cell death induced by co-administration therapy in different cancer cells was positively correlated with pDAPK^S308^ levels [9]. To investigate the correlation between pDAPK^S308^ expression levels and treatment efficiency, we used the IHC method with validated pDAPK^S308^ antibody as a qualitative analysis method. Our results demonstrated that the patients’ samples, which were positive for pDAPK^S308^ in IHC (Figure 2A), are correlated with their duration of treatment (260.6 ± 112.5 vs. 81.4 ± 55.6, *p* = 0.01; Figure 2B).

### 3.6. Salt Form of GW5074 and the Therapeutic Efficiency

The PK data from this study indicated that MG005 has a significantly shorter half-life when compared to sorafenib. The PK data showed that MG005 has a low bioavailability, which limits further evaluation of its efficacy at a higher dosage. Meanwhile, MG005 was also found to have poor absorption, as increasing the dosage from 750 mg QD to 1500 mg QD did not result in a significant increase in the bioavailability of GW5074 in patients. To overcome the challenge of poor bioavailability and to further evaluate the efficacy of MG005, a new oral formulation, i.e., MG010, has been developed (Figure 3A). MG010 showed a significant improvement in solubility compared to MG005, the non-salted GW5074 (17.6:1 at 60 min; Figure 3B).

The in vitro therapeutic effect of MG010 was also checked. When treating the cancer cells with MG010 alone, it has the same outcome as we found by using MG005 (Figure 3C,D; *p* = 0.5 at 24 h, *p* = 0.99 at 48 h). Moreover, consistent with the anti-cancer effect in MG005, MG010 co-administration with sorafenib was also found to likewise suppress the growth of cancer cells (Figure 3C,D; *p* = 0.99 at 24 h, *p* = 0.5 at 48 h).

## 4. Discussion

This phase I, open-label, non-randomized, single-center, dose-defining escalation study with three dose cohorts was designed to assess the safety and anti-tumor activity of GW5074 (MG005) in combination with sorafenib in patients with advanced or metastatic solid tumors who were refractory to/intolerant of currently available therapies and with ECOG performance status score of ≤2. Although GW5074 and sorafenib are both Raf inhibitors, their use in combination provides a novel mechanism of anti-tumor activity targeting cancer cell necroptosis induced by mitochondrial dysfunction, which is distinct from current Raf inhibitor therapies. For our population (patients with histologically confirmed solid tumors relapsed after and/or refractory to standard therapy, or intolerable to marketed available treatments) in the trial, there is a highly unmet medical need for treatment options, and patients are able to benefit from MG005 and sorafenib use in combination due to the expected potency and low toxicity.

The safety data showed that the combination of MG005 and sorafenib was generally well tolerated, with an acceptable and manageable AE profile. Sorafenib has been approved for the treatment of advanced renal cell carcinoma, hepatocellular carcinoma, and thyroid cancer. The most common adverse reactions, including fatigue, weight loss, rash/desquamation, hand–foot skin reaction, alopecia, nausea, anorexia, diarrhea, and abdominal pain, were observed and considered tolerable to patients receiving the therapy. Our dosage of sorafenib (200 mg taken daily) for use in combination with MG005 is much lower than the 800 mg of the highest daily dose approved in the past (2 × 200 mg tablets taken twice daily). Pre-clinical repeated toxicity studies conclude that MG005 did not cause major abnormality except emesis and watery stool, regarded as non-toxic effects due to a lack of toxicological evidence from pathology and histopathology examinations [9]. The no-observed-adverse-effect-levels (NOAELs) in the studies were therefore defined as the highest dose levels tested in Sprague Dawley Rats and Beagle dogs. MG005 is neither mutagenic nor genotoxic according to the results of the employed in vitro and in vivo studies [9]. In addition, our previous findings showed that the cell death induced by a combination of GW5074 and sorafenib depends on the presence of pDAPK^S308^ [9]. Compared to various cancer cell lines and renal carcinoma, the expression level of pDAPK^S308^ was found to be relatively low or undetected in normal cells/tissues, indicating that the risk of the off-target toxicity is low. Compared with AEs reported with the use of sorafenib, only two grade ≥3 AEs were seen in patients treated with this regimen. Longer-term treatment of patients at our final dosage suggested that most could continue treatment with dose reductions to manage treatment related adverse events (TRAEs). Subcostal pain was more likely to be attributed to complications of progressive cancer. One patient had liver function impairment (8.3%), which was self-limited. Grade 3 pneumonia was also reported in a patient from Cohort 2, which was unexpected, and the cause is still unidentified. Further investigation and safety monitoring for AE regarding infections may be needed.

In this study, we demonstrated that a higher dose of MG005 might interfere with the PK and metabolism of sorafenib. Previous publications have indicated that sorafenib is metabolized primarily in the liver, undergoing oxidative metabolism mediated by CYP3A4 [17,18], which is also found to be inhibited by GW5074 in vitro according to the results of CYP450 enzyme assays. Thus, this could be the potential mechanism of these observation results. Moreover, we found that increasing the frequency of GW5074 treatment can increase the drug concentration in circulation. Dividing dosages will be a better choice for taking GW5074 in future trials. In the study, the pharmacokinetics of MG005 do not exhibit a large variability after increasing dosage. It also shows that increasing the dosage of MG005 cannot increase the drug concentration in circulation. After oral administration, GW5074 is absorbed relatively slowly, probably due to poor solubility. Our new salted form of GW5074, MG010, in contrast, has better solubility, and it also provides improved therapeutic efficacy. Sorafenib and MG010 seem to have no pharmacokinetic interactions when administered in combination. Thus, an ongoing phase 2 trial with combination therapy will be initiated for the population with advanced tumors.

DAPK is an important tumor suppressor kinase involved in apoptosis and autophagy. Auto-phosphorylation at Ser-308 inhibits its catalytic activity [9]. The contribution of pDAPK^S308^ molecules towards the progression of solid cancer has not yet been clarified, in part because of a lack of a standardized method to evaluate pDAPK^S308^ expression. Therefore, we developed a novel method for the evaluation of pDAPK^S308^ expression in solid cancer cells and examined its association with clinic-pathological characteristics. This IHC method enabled us to conduct a qualitative analysis which can be a biomarker as a predictor of treatment efficiency. Our findings suggest that the pDAPK^S308^ expression in the primary tumor can be a predictor of pDAPK^S308^ expression in the metastatic tumor, which can help us to avoid unnecessary invasive diagnostic procedures. Intriguingly, the pDAPK^S308^ expression on CTCs retrieved from the patients is also correlated with the patients’ DOR. Thus, this can be a potential biomarker for predicting treatment response in our ongoing trial. Further biomarker studies are required in phase 2 studies to allow us to draw conclusions on clinical usefulness.

Despite recent progress in molecular biology and the development of novel anticancer therapeutic agents, treatment outcomes in most advanced cancers remain poor. The diversity of molecular abnormalities partly contributes to the resistance to therapy. Hence, developing combinations of anticancer drugs that exhibit synergistic activities seems to be a practical strategy [19]. Nevertheless, in clinical trials, there are still many questions that need to be addressed for effective combination of targeted agents [20,21]. For instance, there are challenges in the attribution of anti-tumor activity and toxicity to the individual agents vs. the combination effect. Cumulative toxicity, pharmacodynamic, and pharmacokinetic end points should be considered to better evaluate combined therapies in clinical practice. In conclusion, it appears that dual treatment with the Raf inhibitors sorafenib and GW5074 will not lead to an undesirable toxicity profile. Furthermore, there is a convincing anti-tumor effect in the population of advanced cancers. This finding is encouraging and may launch evaluation opportunities in future practice. Our new salted form of GW5074, MG010, will be more effective and will be used in our ongoing trial.

## Figures and Tables

**Figure 1 jcm-11-02183-f001:**
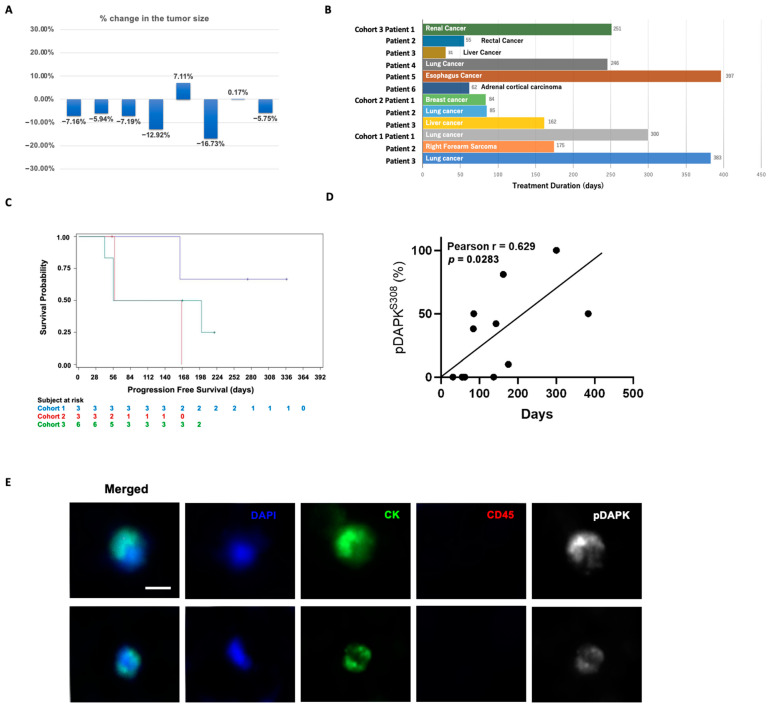
(**A**) Waterfall diagram showing changes in tumor size in target lesions in eight patients who achieved SD. (**B**) Treatment duration of each patient. (**C**) Overall survival of the patients. (**D**) Correlation of proportion of pDAPK + CTC numbers with treatment duration. (**E**) Immunofluorescence staining of representative circulating tumor cells obtained from patients. Cancer cells fulfilled criteria for CTCs, including: CK−positive (green), CD45−negative cells (non-red), and nucleated (blue). Scale bar = 5 µm.

**Figure 2 jcm-11-02183-f002:**
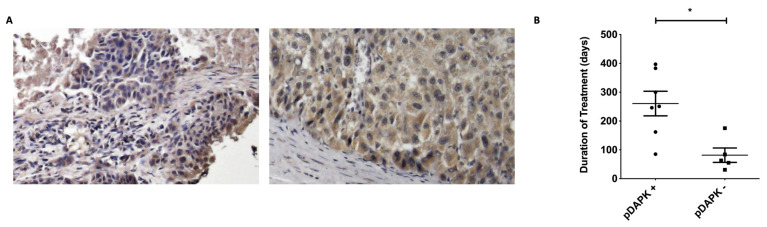
(**A**) Representative patient’s primary tumor IHC of pDAPK^S308^ positive staining. Left, renal cell carcinoma. Right, hepatic cell carcinoma. 400×. (**B**) Association of duration of treatment and pDAPK staining status. * *p* = 0.01.

**Figure 3 jcm-11-02183-f003:**
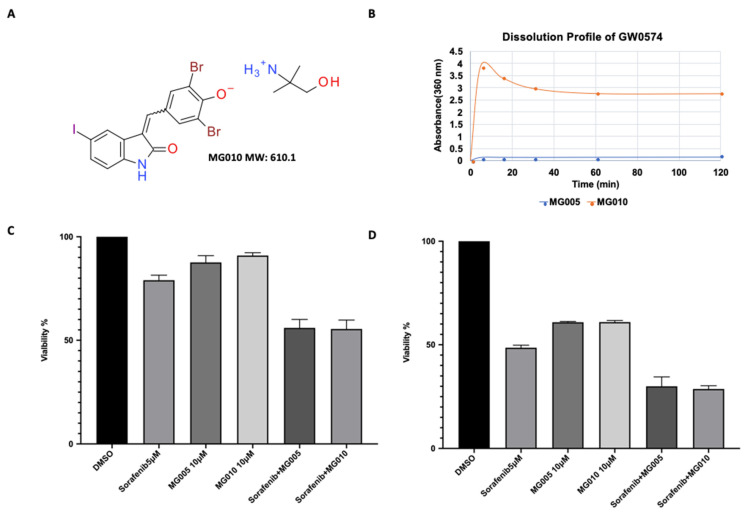
(**A**) Chemical structure of MG010. (**B**) Solubility of MG005 and MG010. (**C**) Viability test 24 h after ACHN treated with different drugs. (**D**) Viability test 48 h after ACHN treated with different drugs.

**Table 1 jcm-11-02183-t001:** Patient characteristics.

	Patients (*n* = 12), *n* (%)
Age (y)	
Median	56.6
Range	25–68
Gender	
Female	7 (58.3)
Male	5 (41.7)
ECOG performance status	
0	
1	
2	
Type of tumor	
Hepatocellular cancer	2 (16.7)
Adenocarcinoma of lung	4 (33.3)
Breast cancer	1 (8.3)
Adrenal cortical carcinoma	1 (8.3)
Esophageal cancer	1 (8.3)
Adenocarcinoma of rectum	1 (8.3)
Renal cell carcinoma	1 (8.3)
Right forearm sarcoma	1 (8.3)
Prior treatment	
Chemotherapy	12 (100)
Radiotherapy	2 (16.7)
Target therapy	8 (66.7)
Immuno-checkpoint inhibitor therapy	1 (8.3)

**Table 2 jcm-11-02183-t002:** Adverse events at least possibly related to study treatment.

Adverse Event	No. (%)		
	Grade 1	Grade 2	Grade 3
Any	7 (58.3)	1 (8.3)	2 (16.7)
General symptoms			
Fatigue	0	0	0
Weight loss	3 (25)	0	0
Dizziness	1 (8.3)	0	0
Hypertension	1 (8.3)	1 (8.3)	0
Insomnia	1 (8.3)	0	0
Gastrointestinal			
Mucositis	1 (8.3)	0	0
Abdominal pain	2 (16.7)	0	0
Nausea	1 (8.3)	0	0
Vomiting	2 (16.7)	0	0
Diarrhea	0	0	0
Constipation	0	0	0
GERD	1 (8.3)	0	0
Erosive gastritis	1 (8.3)	0	0
Duodenal ulcer	1 (8.3)	0	0
Dermatologic			
Rash (desquamation)	0	0	0
Rash (acneiform)	1 (8.3)	0	0
Dry skin	0	0	0
Pruritis	1 (8.3)	0	0
Hand and foot syndrome	1 (8.3)	0	0
Hyperkeratosis	2 (16.7)	0	0
Papule	1 (8.3)	0	0
Eczema	2 (16.7)	0	0
Hematologic			
Anemia	0	0	0
Leukopenia	1 (8.3)	0	0
Thrombocytopenia	0	0	0
Neutropenia	0	0	0
Musculoskeletal			
Back pain	1 (8.3)	0	0
Myalgia	0	0	0
Sacroiliitis	1 (8.3)	0	0
Subcostal pain	0	0	1 (8.3)
Laboratory test results			
AST increased	1 (8.3)	0	0
ALT increased	1 (8.3)	0	0
Hyponatremia	0	0	0
Hypokalemia	0	0	0
Hyperkalemia	0	0	0
Hypertriceridemia	1 (8.3)	0	0
Others			
Conjunctivitis	1 (8.3)	0	0
Gynecomastia	1 (8.3)	0	0
Cough	2 (16.7)	0	0
Pneumonia	0	0	1 (8.3)
Acute bronchitis	1 (8.3)	0	0
Vocal cord paralysis	1 (8.3)	0	0
Hemorrhoids bleeding	1 (8.3)	0	0
Dental caries	1 (8.3)	0	0
Hydrocephalus	1 (8.3)	0	0

Abbreviations: ALT, alanine aminotransferase; AST, aspartate aminotransferase. No AEs were found in five patients.

**Table 3 jcm-11-02183-t003:** Sorafenib and GW5074 pharmacokinetic variables.

**A. Sorafenib Pharmacokinetic Variables While Co-Administered (Mean Values and Range) by Dose Level**				
		**Cohort 1**	**Cohort 2**	**Cohort 3**
Day 28	C_max,ss_ (ng/mL)	6043.4 ± 2615.4	3726.5 ± 1533.7	2930.6 ± 1788.1
	T_max,ss_ (Hr)	6.0 (2.0–12.0)	8.0 (2.1–23.5)	2.0 (1.0–4.1)
	AUC_0→τ,ss._ (h·ng/mL)	88,597.1 ± 37,755.1	51,423.1 ± 17,695.1	30,625.3 ± 16,486.7
**B. GW5074 Pharmacokinetic Variables While Co-Administered (Mean Values and Range) by Dose Level**				
		**Cohort 1 (*n* = 3)**	**Cohort 2 (*n* = 3)**	**Cohort 3 (*n* = 3)**
Day 28	C_max,ss_ (ng/mL)	685.6 ± 335.4	786.0 ± 180.0	620.0 ± 382.6
	T_max,ss_ (Hr)	6.0 (5.0–6.0)	4.0 (2.1–4.1)	14.0 (0.0–23.9)
	AUC_0__→τ,ss._ (h·ng/mL)	5639.9 ± 2675.9	5553.7 ± 950.2	6249.6 ± 2619.6

All data are mean ± SD with the exception of T_max,ss_, which is median (range). C_max,ss_, peak plasma concentration at steady state; T_max,ss_, time to reach peak concentration at steady state; AUC_τ,ss_, area under the concentration–time curve within one dosing interval.

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
