# Peer review of "Phase I Targeted Combination Trial of Sorafenib and GW5074 in Patients with Advanced Refractory Solid Tumors"

_jcm, 2022, doi:10.3390/jcm11082183_

Round 1

Reviewer 1 Report

In this study, Kao et al. have investigated the safety, efficacy, and pharmacokinetics of a combination therapy consisting of two Raf inhibitors, namely sorafenib, and GW5074 in twelve solid cancer patients. Below are a few suggestions the authors may consider to improve their manuscript:

  • Since this combination approach employs two inhibitors against a common target, the relevance and rationale of using such an approach need to be emphasized in the text. 
  • The source of GW5074 and sorafenib should be provided in the methods section.
  • In some figures, such as 1C and 3C-F, the axes titles and figure labels are not legible. 
  • There are some typos and grammatical errors that should be rectified.
  • In line 287, it should be 'tumor relapsed' instead of 'tumor elapsed'
  • Lines 299-304, lack a suitable reference. 
  • Line 312; the full form of TRAEs should be stated at the site of first use.

Reviewer 2 Report

Dear authors please find my comments

1) Authors should be very cautious with their results based on this very small number of patients (eg PKs and interaction with sorafenib: what is the statistical power of the reductions described? ; and mainly with efficacy: abstract cites the combination whoed antitumor efficacy: as reviewer sees these results there is no any possibility to support that the combo has antitumor activity)

2)please define better what is the dose escalation scheme and use some references. 

3) can you explain why the DLT observation period was longer for the low dose? and not for the higher ones? (4 vs 8 wks?)

4) the PK animal study is not clearly related with the current phase I trial and should not be reported here

5) was pneumonia gr3 not DR

6) what is the evidence to reduce the standard dose of sorafenib in order to combine it with a drug that has not known activity?
